# Just another bike? Modelling the interdependence of conventional and electric bicycle ownership and the influence of topography using large-scale travel survey data from Germany

Leonard Arning *, Heather Kaths

Chair of Bicycle Traffic, Center for Mobility and Transport, University of Wuppertal, Wuppertal, North Rhine-Westphalia, Germany

* arning@uni-wuppertal.de

## Abstract

Electrical bicycle ownership rates are growing rapidly. Despite differences to conventional cycling, the two types of bicycles are generally not differentiated in travel demand modelling practice. This article analyses the choices to own electric and conventional bicycles in Germany at the personal level. We use data from the "Mobility in Germany" survey and other sources and estimate both a nested logit model and a multivariate probit model. While the average gradient of terrain near the residence has an expected, strong negative influence on the ownership of conventional bicycles, electric bicycle ownership is much less negatively affected. The effect of socio-demographic variables is largely in line with that of the existing literature. A negative correlation of the error terms in the probit model indicates a substitutive relationship between the two ownership decisions. The high nest parameter value in the nested logit model indicates that the decision to own a conventional bicycle is secondary to the decision to own an electric bicycle. The results contribute to a better understanding of the motivations for or against bicycle ownership and create a basis for better consideration of electrical bicycle traffic in transport models.

## Introduction

Between 2012 and 2023, the number of electric bicycles in Germany increased from 1.3 to 9.8 million [1]. By 2023, they already accounted for more than half of newly sold bicycles in the country [2]. Despite this dynamic growth (Fig 1) and the meaningful differences between electric and conventional cycling, most notably concerning speed, user groups, trip purposes, overcoming hills, and trip lengths, there are still few integrated transport models that take into account the effects of the electrification of cycling and none in which e-bikes are considered as a fully-fledged and

**Data availability statement:** The dataset underlying the research presented in the study is the local dataset package B3 of the Mobility in Germany 2017 survey. The authors cannot share this data because it is owned by a third party. However, it is available to other researchers who fulfil the conditions directly from the owner. Namely, the German Federal Ministry for Digital and Transport, represented by the German Federal Highway and Transport Research Institute's "MobilityData-Campus", has made the data available at request under the conditions of signing a user agreement and paying a processing fee of € 100 (plus 19% value-added tax), as was the case for this study. The authors of this study did not receive any special privileges in accessing the data. More information about the dataset and the requirements for access are available from the German Federal Highway and Transport Research Institute's "MobilityData-Campus": https://www.bast.de/DE/Verkehrssicherheit/Fachthemen/MobilityDataCampus/Datenangebot/Datenangebot_node.html. The application form for accessing the data is available at request by contacting mobilitydata-campus@bast.de.''

**Funding:** This research was supported via funding from the German Federal Ministry of Digital and Transport's Bicycle Traffic Endowed Professorship at the University of Wuppertal.

**Competing interests:** The authors have declared that no competing interests exist.

independent means of transport across all modelling stages. This neglect of electric bicycles is partly due to a lack of data and understanding about electric bicycle traffic choice behaviour, which might result in uncertainties regarding the accuracy and forecasting ability of existing models [3].

Differences between conventional and electric bicycles (c-bikes and e-bikes) have been considered in research, particularly with regard to mode and route choice. In reality, however, the choice of whether to travel by c-bike or e-bike is usually preceded by the decision of what type(s) of bicycle to own. C-bike and e-bike ownership should therefore be taken into account when modelling mode choice. This is particularly relevant because the purchase of an e-bike is a more critical decision than the purchase of a c-bike due to the higher investment costs. To be able to analyse and forecast c-bike and e-bike ownership, current bicycle ownership must be examined in detail and modelling approaches must be developed. This study makes such a contribution to representing the diversity of cycling in transport models in a more differentiated way by presenting two models for the combined ownership choice of c-bikes and e-bikes. In particular, we are the first to use discrete choice models to investigate both c-bike and e-bike ownership and to consider average gradient, allowing for insights into how topography affects the two ownership decisions and how they influence each other. Therefore, the following research questions take centre stage:

1. Which factors influence the choice to own a c-bike and/or an e-bike?

2. What role does topography play in particular?

3. How are the two choices interlinked?

The rest of this paper is structured as follows: in section 2 we give an overview of factors influencing the ownership of e-bikes as well as types of discrete choice models that are commonly used for modelling the ownership of mobility tools. Sections 3 and 4 describe the data used to estimate the models and the model specifications. In section 5, we present and interpret the estimated model parameters and discuss

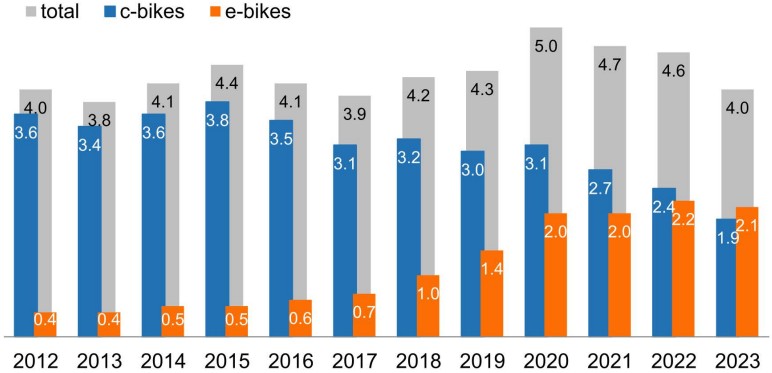

**Fig 1. Development of yearly bicycle sales [mln. ]** in Germany based on data from [2].

shortcomings, further research needs and implications for modelling practice, before ending with our main conclusions in section 6.

## Literature

### Influencing factors on e-bike ownership and use

There is comparatively little research investigating influencing factors on e-bike ownership. Socio-demographic factors were most commonly found to have a major influence on whether someone owns an e-bike, with different user groups demonstrating distinct user behaviours. Table 1 provides an overview of the findings from researchers in some European and North American countries. South and East Asia, where the term "e-bike" is generally used to refer to motorbike-like vehicles instead of bicycles [4], are not considered here.

The nearly unanimous finding that in particular older people own e-bikes suggests that the main motivation for their purchase is to be able to continue cycling despite advancing age and declining fitness. This is consistent with the results of direct surveys on purchase motivation [12]. In contexts with low subjective road safety, cyclists also state that they feel like they can compensate for deficiencies in the infrastructure and differences in speed compared to motorised traffic by riding an e-bike instead of a c-bike [11,12]. It is well established that personal attitudes such as environmental awareness or enthusiasm for cycling are of high relevance to both ownership and use of e-bikes [5,13–15].

Research investigating attitudes towards e-bike use and purchase intentions provides valuable indications of further influencing factors on e-bike ownership. Awareness of e-bikes is a precondition to acquisition. For university employees in California, Handy and Fitch [16] find that after the introduction of an e-bike sharing system, awareness of e-bikes increases substantially and the intention to use an e-bike for commuting increases slightly. In a Norwegian survey, Simsekoglu and Klöckner [17] find that besides socio-demographic factors such as age, purchase intention is also influenced by respondents' awareness of e-bikes, their perceived benefits, as well as subjective and descriptive norms, i.e., whether they believe that others expect them to own an e-bike and that other people own e-bikes. Kaplan et al. [18] report that the intention to use an e-bike in a c-bike and e-bike sharing system is stronger for women and the elderly in Poland. Human needs according to the ERG (existence, relatedness, growth) theory of needs were also found to be important determinants of usage intention, with growth needs relating to a stronger intention to use a c-bike and a weaker intention to use an e-bike. For Polish society overall, Kwiatowski et al. [19] find that public perception of e-bikes is mostly critical; respondents view them as expensive, advantageous only for the elderly, and are largely unaware of other e-bike benefits. Plazier et al. [20] investigate current and potential e-bike use in a rural region of the Netherlands. They find e-bikes are "used among a broad population of varied ages and backgrounds and for different purposes" (p. 1449), that e-bikes likely complement car and substitute c-bike ownership, and that personal attitudes towards safety, fun and health benefits of e-bikes are important determinants of e-bike use.

**Table 1. Literature overview about influencing factors on e-bike ownership.**

| Country, source | Personal traits supporting e-bike ownership | Associated trip purpose |
|---|---|---|
| Denmark [5] | Older age and high income, female, high cycling affinity | Leisure, pick-up and drop-off |
| Germany [6,7] | Older age, middle or high economic status, outside of large cities | Leisure |
| The Netherlands [8] | Older age and high income, female | |
| The Netherlands [9] | Older age | Leisure |
| | Middle-aged, full-time employed | Commute |
| | Middle-aged, part-time employed, female | Leisure, shopping |
| Switzerland [10] | Older age, female, suburban and rural, couples with children, very high and very low income | Commute |
| US and Canada [11] | White, male, older age, high level of education | Leisure |

The role of topography with regards to cycling and the potential of e-bikes is frequently discussed, however little research on its influence on c-bike and e-bike ownership exists. An earlier work already demonstrated a negative correlation between varied topography and bicycle ownership and use in Germany [6]. In a North American survey, "Because I live or work in a hilly area" was the most frequently cited reason for purchasing an e-bike [11]. Such findings lead to the hypothesis that e-bikes are particularly attractive in hilly areas where they can mitigate the negative impact of the topography on cycling. On the other hand, there is evidence from other North American studies that hilliness might have only a small [21] or even insignificant [22] impact on (mostly conventional) bicycle use, both on the level of metro areas and persons. The influence of topography on e-bike ownership therefore remains unclear. We are unaware of any studies on discrete choice models that take into account the topography near the residential location on e-bike ownership. This may be because countries with a pronounced cycling culture and corresponding data are generally comparatively flat. This study closes this research gap.

## Types of discrete choice models for mobility tool ownership

The decisions of individuals or households about whether to own a specific mobility tool is a discrete choice. The utility trade-offs can be described with discrete choice models and the model parameters can be estimated using revealed choice or stated choice data. Past work on mobility tool ownership has focussed primarily on cars and, to a lesser extent, on public transport season tickets [23]. Little attention has been paid so far to modelling bicycle ownership, as the purchase cost of a c-bike is comparatively low and, at least in many European contexts, it can be assumed that every person who is able and willing to ride a c-bike has access to one. The higher purchase cost of an e-bike and the specific motivators for use increase the need for more differentiated modelling of the availability of bicycles.

Logit models are the most common model type for mobility tool ownership. The estimation of separate, binary logit models for each mobility tool would be inaccurate, as the decisions on their ownership are made dependently. Therefore, multinomial logit models are used that formulate choice options that consist of combinations of different mobility tools (bundles). Fatmi et al. [23] apply such a model to study mobility tool ownership of young adults in Toronto. Kohlrautz and Kuhnimhof [7] apply a similar approach to data from the German MiD 2017 survey to understand bicycle ownership as well as c-bike and e-bike mode choice, however without differentiating between c-bikes and e-bikes in ownership modelling or taking into account topography.

Multinomial logit models inherently assume the independence from irrelevant alternatives (IIA) property, which may not hold when dealing with bundles of choice options. Nested and cross-nested logit models provide a solution by allowing for correlations among related alternatives. Bundles of mobility tools are placed within nests (cross-nested logit allowing for overlapping nests), with each nesting level representing the decision about one mobility tool. Püschel et al. [24] use both a nested and cross-nested logit as well as a machine learning model to investigate car, car sharing and public transport season ticket ownership of residents of Hamburg, Germany. Handy et al. [13] employ a nested logit model to jointly investigate bicycle ownership and consequent use by residents of six small US cities. On the top level a decision between "has no bike" and "has bike(s)" is made, and within the latter, a nested choice between "bikes non-regularly", "regular transportation-oriented bicyclist", and "regular non-transportation-oriented bicyclist" is made.

Probit models are widely applied in studies of mobility tool ownership due to their ability to account for interdependencies among choices by modelling correlations between error terms as explicit parameters. For example, individuals holding a public transportation season ticket are likely to have a lower utility for (additional) car ownership, and vice versa. In contrast to the previously mentioned approaches, studies employing multivariate probit models specify utility functions for individual mobility tools rather than a bundled set of tools, enabling more intuitive interpretation of parameters associated with each choice. Becker et al. [25] use such an approach to model the ownership of cars, public transport season tickets and car-sharing services in Switzerland. Scott and Axhausen [26] introduce the ordered probit model to model the number of public transport season tickets and cars per household in Switzerland. Yamamoto [27] uses a trivariate binary

probit model to compare factors influencing the ownership of bicycles, motorbikes and cars in Osaka and Kuala Lumpur. Ma et al. [28] apply a multivariate ordered probit model to investigate car, motorcycle, e-bike, and c-bike ownership of households in Hangzhou, China.

At the household level, it is sensible to quantify the number of available mobility tools. This can be achieved with an ordered logit approach. Here, while a single utility function is estimated for each mobility tool, threshold values indicating when a household owns an additional mobility tool (e.g., two cars instead of one) are also estimated. Maltha et al. [29] use this approach to model car ownership in the Netherlands. Pinjar et al. [14] combine an ordered logit model for the number of bicycles owned by a household with a binary logit model for the household's choice of residing in a bicycle-friendly neighbour-hood in a joint model system. This allows for residential self-selection effects to vary across households. Zhang et al. [30] use a zero-inflated Poisson model to investigate e-bike ownership in Zhonshan, China. It consists of a binary logit model aimed at predicting whether a household owns an e-bike at all, followed by a Poisson model predicting the number of e-bikes owned by households that own one or more e-bikes. Ding et al. [4] expand on this work by applying a semi-parametric generalized additive mixed model to the data, which allows for more relaxed assumptions regarding the linearity of the variables.

In contrast to static modelling approaches, dynamic approaches describe the change in ownership over time instead of the momentary stock of mobility tools in a household. For example, Gu et al. [31] investigate the influence of life course events (moving, birth of a child, etc.) on the change in the ownership of a car using an error component random parameter logit model in which the constants of the utility functions are household-specific and normally distributed. The choice options here consist of combinations of buying or keeping a car as well as the purchase of additional sustainable mobility tools.

## Materials and methods

### Data

This retrospective study is based on household and person-level data from the B3 local dataset package of the "Mobility in Germany 2017" (German: "Mobilität in Deutschland", MiD 2017) survey [32] and two additional spatial datasets. The data is anonymized, does not contain medical information, and is publically available from the German Aerospace Center. For this reason, we did not seek approval from an ethics committee. In the MiD, the availability of c-bikes and e-bikes is recorded at the person-level and can assume different values for different people in the same household. For example, survey respondents frequently indicated no e-bike availability for underage household members, even when an e-bike was available to other household members. The socio-demographic variables age, level of education, gender and occupation are also available at the person-level. The variables economic status, household size and grid cell of the place of residence are recorded at the household-level, but are also treated at the person-level in our models for the sake of uniformity. Below, we describe our data processing. The respective source code is available on GitHub: https://github.com/buw-bicycle-traffic/ebike-ownership-model.

The spatial variables "spatial typology" (German: "Raumtyp", degree of urbanisation) and "gradient" were linked to the MiD person-level data using the residential location which is coded in the MiD using a standardised grid of 1-by-1-km large cells [33]. The spatial typology was included as there are clear differences between the use of c-bikes and e-bikes in urban and rural areas in Germany [6]. Spatial typology is defined at the municipality level in the RegioStaR dataset [34], but neither the persons nor the 1km grid cells are assigned to municipalities in the MiD dataset. For the corresponding 250-by-250-m grid cells, however, a bridge between cells and municipalities is available. Therefore, for the sake of simplicity, each 1km grid cell was assigned one 250m grid cell located in its centre (more precisely, southwest of the centre of the 1km grid cell) in order to be able to assign a spatial typology code to each person via the grid cells and the official municipality key. Fig 2 shows the spatial typology as assigned to the grid cells.

The variable gradient is based on a topographic dataset provided by Burgdorf and Pütz [35]. For every 250-by-250-m large grid cell, it records the average gradient of terrain across that grid cell and its eight surrounding neighbours. We aggregate this further by computing the average gradient of each 1km cell based on its sixteen constituent 250m cells.

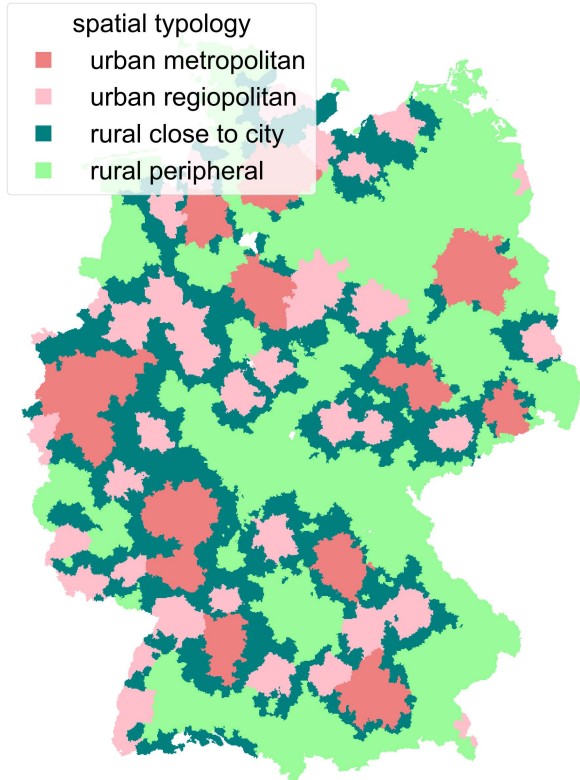

**Fig 2. Spatial typology of 1km grid cells.** Grid cell position from [33] and spatial typology of grid cells based on [34], both under DL-DE->BY-2.0 license.

Even though most bicycle trips can be expected to reach beyond this immediate vicinity around the residential location, testing showed that further increasing the area used for computing individuals' gradient values decreased model fit. The resulting gradient values assigned to the grid cells are shown in Fig 3.

All observations for which not all variables were fully recorded were excluded. Most notably, there was no information on bicycle availability for 26% of all respondents. Due to correlation between the youngest age group and the lowest level of education, we interact age with level of education and omit the lowest level of education from the utility functions in addition to the reference category "Abitur". A low number of cases of adults with no education therefore also had to be removed. As the variables spatial typology and gradient require spatial localisation, only persons for whom the residential location was recorded at least at the 1km grid cell level were considered. This data processing reduces the available sample size from 316,361 (raw data) to 161,963 persons. Due to high computational demands of a probit model, a random subsample of 30,000 persons was used for model estimation. This sample size ensured a balance between computational efficiency and model reliability. Table 2 describes the statistical distribution of the categorical variables in the original raw data and in the sample used for model estimation. Fig 4 shows the spread of the continuous variable gradient for the estimation sample as a box plot. Since all previous works identified age as an important influencing factor on e-bike ownership, Fig 5 visualises the shares of bicycle ownership across age groups.

Only in one case there is a strong correlation (in its amount larger than 0.60) between independent variables of different groups. This is the case for "age 0-17" x "education 'no qualification (yet)'" (0.80). To address this, we interact those variables (see section 4). We do not include other mobility tools as explanatory variables for bicycle ownership because

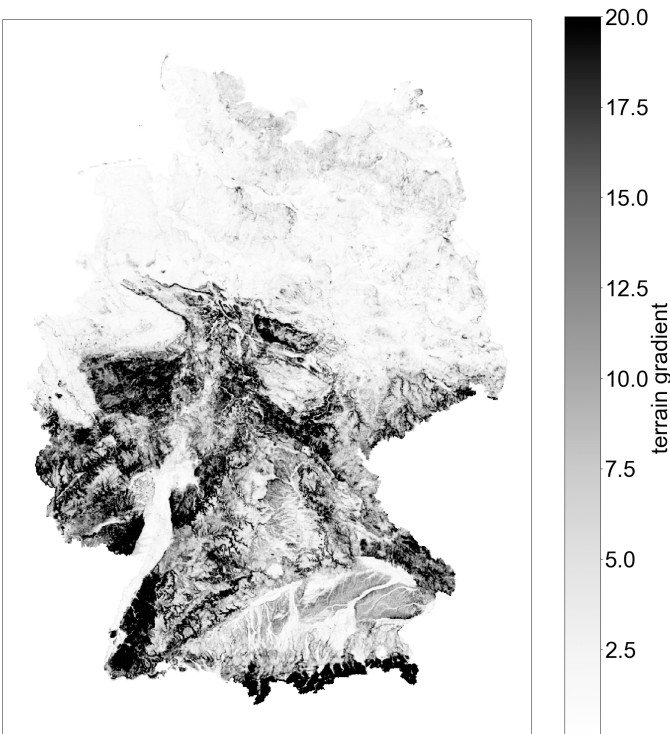

**Fig 3. Average terrain gradient [%] of 1km grid cells.** Grid cell position from [33] under DL-DE->BY-2.0 license, gradient based on data provided by [35].

of model hygiene: car ownership and transit cards are influenced by similar socio-economic factors as bicycle ownership, which could introduce endogeneity. Without knowing the sequence of these decisions, including them may obscure the interpretation of bicycle ownership determinants. While a comprehensive model could treat all mobility tools as jointly determined, this would add complexity and reduce clarity. Therefore, we focus solely on bicycle ownership in this model.

## Models

Based on findings from the literature, several model variants with analogous utility functions were tested. We report the model specification and results for both a nested logit model and a multivariate probit model. We present two different models because they have distinct advantages: While model parameters of the multivariate probit can be interpreted more intuitively due to its utility functions representing one type of bicycle each instead of bundles, the nested logit allows for the computation of odds ratios and achieves a higher model fit. Furthermore, the nested logit captures the dependency between the two choices by bundling them and accounting for similarities between the bundles using nests, while the multivariate probit does not bundle them but captures the mutual influence as a correlation of the error terms. This allows for different perspectives on the nature of the two choices' relatedness.

The Python package Biogeme 3.2.10 [36] was for the logit model, while the R package mvProbit 0.1–10 [37] was used for the probit model. Like for data processing, the source code for model estimation is available on GitHub.

## Nested logit

Our nested logit model assumes that each person decides in favour of one of four possible bundles $b$ of bicycle types. These bundles consist of either only a c-bike ($b = 1$), only an e-bike ($b = 2$), both types ($b = 3$), or no bicycle at all ($b = 4$). According to Equation 1, each person chooses (dependent variable $Y$) the option that is associated with the highest utility $U$.

**Table 2. Descriptive statistics of variables at person-level.**

| Variable and level | Raw data [%] | Estimation sample [%] | Variable and level | Raw data [%] | Estimation sample [%] |
|---|---|---|---|---|---|
| Bicycle ownership | | | Household size | | |
| 1 - only c-bike | 72.9 | 73.1 | 1 - 1 person | 11.4 | 16.8 |
| 2 - only e-bike | 3.0 | 2.9 | 2 - 2 persons | 42.3 | 48.8 |
| 3 - both | 5.1 | 4.7 | 3 - 3 person | 17.7 | 15.6 |
| 4 - neither | 18.7 | 19.3 | 4 - 4 persons or more | 28.6 | 18.9 |
| Age | | | Occupation | | |
| 1 - 0-17 | 12.8 | 2.7 | 1 - employed | 45.8 | 49.3 |
| 2 - 18-29 | 9.4 | 9.3 | 2 - education | 14.8 | 7.9 |
| 3 - 30-39 | 8.0 | 9.3 | 3 - domestic | 3.7 | 3.7 |
| 4 - 40-49 | 12.7 | 13.3 | 4 - retired | 29.3 | 35.8 |
| 5 - 50-59 | 20.0 | 21.4 | 5 - other | 6.3 | 3.3 |
| 6 - 60-69 | 18.1 | 20.5 | Economic status | | |
| 7 - 70-79 | 13.9 | 17.5 | 1 - very low | 3.7 | 3.5 |
| 8 - 80 and older | 4.8 | 6.1 | 2 - low | 8.9 | 8.9 |
| Level of education | | | 3 - middle | 39.3 | 44.0 |
| 1 - none (yet) | 13.7 | 2.7 | 4 - high | 38.1 | 34.1 |
| 2 - "Volks-/Hauptschule" | 16.8 | 17.9 | 5 - very high | 10.0 | 9.5 |
| 3 - "Mittlere Reife" | 23.9 | 25.5 | Spatial typology | | |
| 4 - "Abitur" | 14.9 | 16.9 | 11 - urban metropolitan | N/A | 55.7 |
| 5 - university degree | 28.8 | 34.4 | 12 - urban regiopolitan | N/A | 20.1 |
| 6 - other qualification | 1.9 | 2.2 | 21 - rural close to city | N/A | 12.8 |
| Sex | | | 22 - rural peripheral | N/A | 11.3 |
| 1 - male | 50.3 | 50.1 | | | |
| 2 - female | 49.7 | 49.9 | | | |

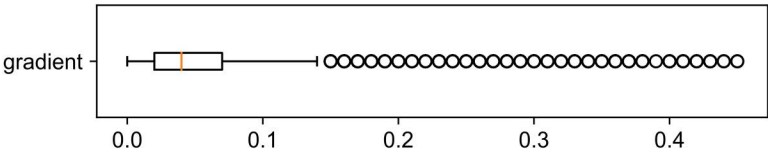

**Fig 4. Boxplot of average gradient near residential location at person-level.**

$$Y = \begin{cases} 1, & \text{if } U_{b=1} = \max(U_b) \\ 2, & \text{if } U_{b=2} = \max(U_b) \\ 3, & \text{if } U_{b=3} = \max(U_b) \\ 4, & \text{if } U_{b=4} = \max(U_b) \end{cases} \quad (1)$$

The utility of the reference bundle 4 (owns neither bicycle) is set to 0. For the other three bundles $b$, the utility $U$ for each person is described by utility functions according to Equation 2. They are identical in structure for each of the four bundles and differ only in the parameter values to be estimated. $V$ is the observable part of utility. The alternative specific constant $ASC$ of every bundle is the same across all persons. $\beta_{b,\ gradient}$ is the bundle-specific parameter for the person-specific

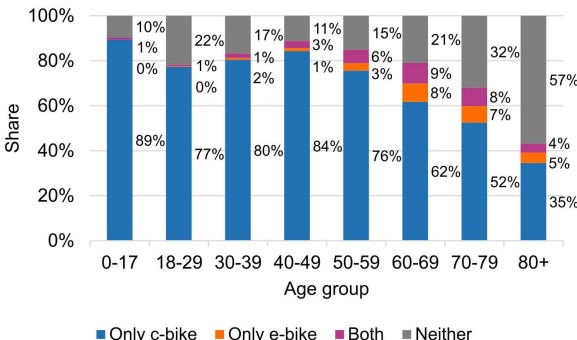

**Fig 5. Bicycle ownership across age groups.**

variable *gradient*. Linking gradient with an additional exponential parameter instead of just a linear parameter was tested but rejected due to the negative impact on the model fit. $\hat{\beta}_{b,spatialtyp}$ and $\widehat{spatialtyp}$ are vectors of the parameters and values respectively of the three dummy variables for spatial typology. $\hat{\beta}_{b,SD}$ and $\widehat{SD}$ represent the same for the socio-demographic dummy variables. The latter is expanded in Equation 3. Note the interactions of age with occupation and level of education. This is because the lowest age category correlates with the occupation "in education" and the level of education "none (yet)". With this specification, parameter values for occupation and level of education are estimated only for adults, while the parameter for the youngest age group captures the combined effect of age and age-typical occupation and level of education for that age group. In addition to the reference category, the lowest level of education was also omitted since it only applies to persons in the youngest age category.

$$U_b = V_b + \varepsilon_b = ASC_b + \beta_{b,gradient} * gradient + \hat{\beta}_{b,spatialtyp} * \widehat{spatialtyp} + \hat{\beta}_{b,SD} * \widehat{SD} + \varepsilon_b \tag{2}$$

$$\widehat{SD} = (age_1, age_2, age_3, age_5, age_6, age_7, age_8, \ edu_2 * (1 - age_1), edu_3 * (1 - age_1), edu_5 * (1 - age_1), edu_6$$
$$* (1 - age_1), \ sex_2, occu_2 * (1 - age_1), occu_3 * (1 - age_1), occu_4 * (1 - age_1), occu_5 * (1 - age_1), \tag{3}$$
$$eco_1, eco_2, eco_4, eco_5, hhsize_2, hhsize_3, hhsize_4)$$

Using the behavioural assumption from Equation (1) and the general utility definition from Equation (2), the probability of choosing alternative $b$ over the other alternatives $b'$ becomes:

$$P(Y = b) = P(V_b + \varepsilon_b > V_{b'} + \varepsilon_{b'} \forall b' \neq b) \tag{4}$$

Assuming Gumbel-distributed error terms, one can derive a closed form for the multinomial logit choice probability, as first demonstrated by McFadden [38]:

$$P(Y = b) = \frac{e^{V_b}}{\sum_{b' \in Y_n} e^{V_{b'}}} \tag{5}$$

In multinomial logit, the error terms $\varepsilon_b$ are assumed to be independent and identically distributed (i.i.d.) between individuals and bundles. That assumption would be problematic in this case because the bundles contain overlapping mobility tools. In nested logit, similar alternatives (i.e., options sharing unobserved attributes) are grouped into nests ($M$). This allows for correlated error terms within each nest but assumes independence between nests. Namely, the error term $\varepsilon_b$ is decomposed into two parts:

$$\varepsilon_b = \xi_n + \eta_b, \tag{6}$$

where $\xi_n$ is the component shared by all alternatives in nest $n$, and $\eta_b$ is the i.i.d. component for bundle $b$. The probability of choosing a specific bundle is the product of the conditional probability of $b$ within its nest $n$ and the probability of selecting that nest:

$$P(Y = b) = P(Y = b \mid M = n) * P(M = n) = \frac{e^{V_b/\mu_n}}{\sum_{b' \in Y_n} e^{V_{b'}/\mu_n}} * \frac{e^{\mu_n \Gamma_n}}{\sum_{n' \in M} e^{\mu_{n'} \Gamma_{n'}}} \tag{7}$$

where $\Gamma_n$, the log-sum term, is given by:

$$\Gamma_n = ln \sum_{b' \in Y_n} e^{V_{b'}/\mu_n} \tag{8}$$

Five nesting structures depicted in Fig 6 were tested. Nesting structure 2 was chosen due to highest adjusted . In the chosen nesting structure, the single nest parameter  determines the degree of similarity between options within this nest, namely owning an e-bike but not owning a c-bike, and owning both an e-bike and a c-bike. A value of 1 implies no correlation, reducing the model to multinomial logit, while higher values indicate increasing similarity among bundles within the nest. For further information on nested logit, we refer to Koppelman and Wen [39].

**Multivariate probit**

In our probit model, a person does not choose one out of four alternatives, but decides in two binary decisions between two alternatives each. These decisions are whether to own a c-bike and whether to own an e-bike. The two dependent variables $Y_t$ describe whether a person owns a bicycle of type $t$ (conventional and electric) according to Equation 9:

$$Y_t = \begin{cases} 1, & \textit{if } U_t > 0 \\ 0, & \textit{else} \end{cases} \tag{9}$$

$U_t$ is the utility of a person to own a specific type of bicycle $t$. Equation 10 describes the structure of these two utility functions, which is identical to the nested logit model in the previous section. However, note the replacement of $b$ by $t$.

$$U_t = V_t + \varepsilon_t =$$

$$ASC_t + \beta_{t,gradient} * gradient + \hat{\beta}_{t,spatialtyp} * \widehat{spatialtyp} + \hat{\beta}_{t,SD} * \widehat{SD} + \varepsilon_t \tag{10}$$

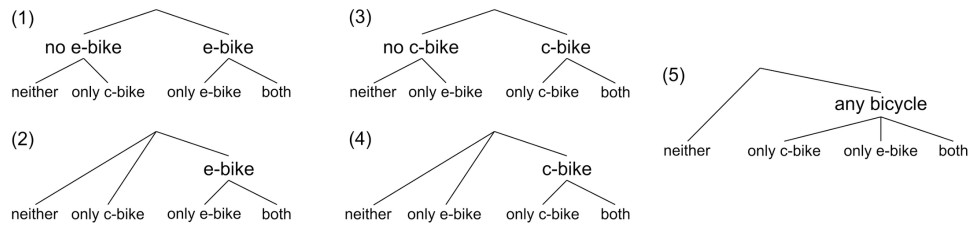

**Fig 6. Tested nesting structures.**

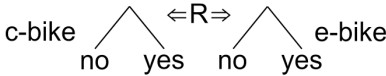

**Fig 7. Decision structure of the multivariate probit model.**

As in the logit model, the error terms $\varepsilon_t$ represent the unobserved part of the utility. However for probit, they are assumed to be normally distributed between the individuals. In order to take into account the mutual influence of the decisions, they are also assumed be correlated for each person between the two decisions. Namely, they follow a bivariate normal distribution:

$$\begin{bmatrix} \varepsilon_{cbike} \\ \varepsilon_{ebike} \end{bmatrix} \sim \mathcal{N}\left( \begin{bmatrix} 0 \\ 0 \end{bmatrix}, \begin{bmatrix} 1 & R \\ R & 1 \end{bmatrix} \right) \tag{11}$$

where the global correlation coefficient $R$ is an additional model parameter that is estimated using the data. The joint probability $P$ that $Y_{t=conv}$ takes the value $y_{t=conv}$ (0 or 1) and $Y_{t=elec}$ takes the value $y_{t=elec}$ (0 or 1) is given by Equation 12:

$$P(Y_{conv} = y_{conv}, Y_{elec} = y_{elec}) =$$

$$\Phi_2[(2y_{conv} - 1) * V_{conv}, \ (2y_{elec} - 1) * V_{elec}, \ (2y_{conv} - 1) * (y_{elec} - 1) * R] \tag{12}$$

Here, is the cumulative density function of the bivariate normal distribution. The correlation captures the mutual influence of the two decisions: If it is positive, unobserved factors increase the likelihood of jointly owning (or not owning) both types of bicycles (i.e., complementary effects), while a negative value of indicates that unobserved factors reduce the likelihood of jointly owning (or not owning) both types of bicycle (i.e., substitutive effects). The model structure is visualized in Fig 7. Note that unlike in the nested logit model, the multivariate probit model considers the decisions about each type of bicycle not hierarchically but separately, being linked by correlated error terms. For further information on multivariate probit, we refer to Greene [40].

## Results and discussion

### Parameter values and model quality

After presenting the model specifications, we now report the results of model estimation. Tables 3 and 4 show the estimated model parameters of the nested logit and the multivariate probit model. Reference categories used for model identification are included in cursive. The choice option "no bicycle owned" is the reference choice option for the nested logit model, with its utility set to 0. For each of the two binary decisions in the multivariate probit model, not owning the respective bicycle type is the reference choice option, with ownership being assumed if the utility for owning that type is larger than 0. All parameters are tested against the null-hypothesis of them being 0, with the exception of the nest parameter, where it is tested against the null-hypothesis of being 1. Table 5 compares the two models. Note that while for probit, model parameters can be compared across bicycle types, with nested logit every bundle contains the outcome of two decisions regarding c-bike and e-bike ownership and one needs to scale using the nest parameters.

ASC: The constants have the expected signs and express the generally higher hurdle (especially price) when buying an e-bike than a c-bike.

Gradient: The average gradient near the residential location has a significant negative influence on the utility of owning a c-bike. In the probit model, an average gradient of 2.8% is as detrimental to owning a c-bike as the fact that a person is above 80 years old (compared to between 40 and 49). Such a gradient value is common in only very moderately hilly areas. A different picture emerges for e-bikes: The gradient parameter in the probit model is larger than 0, meaning gradient has a positive impact on e-bike ownership. In the nested logit model, the difference between the two types of bicycles appears less extreme at first glance, however the difference in utility between the nested bundles "c-bike and e-bike" and "only e-bike" is scaled by the value of the nest parameter.

Urban vs. rural: The overall picture that emerges from the nested logit model regarding spatial typology is that for rural residential locations, there is a higher utility for an e-bike, but with no clear indications for how it affects c-bike ownership. The probit model allows for a more differentiated picture with regard to e-bikes: Compared to the reference

**Table 3. Parameter values for the nested logit model.**

| Parameter | Only c-bike | | | C-bike and e-bike | | | Only e-bike | | |
|---|---|---|---|---|---|---|---|---|---|
| | Value | Rob. p-val. | Sig. | Value | Rob. p-val. | Sig. | Value | Rob. p-val. | Sig. |
| constant | 1.99 | 0.000 | *** | -1.56 | 0.000 | *** | -1.71 | 0.000 | *** |
| gradient | -5.99 | 0.000 | *** | -3.48 | 0.000 | *** | -3.19 | 0.000 | *** |
| *spat. typ. metrop. urban* | | | | | | | | | |
| spat. typ. regiop. urban | -0.089 | 0.030 | * | 0.112 | 0.092 | * | 0.128 | 0.058 | * |
| spat. typ. rural close to city | -0.021 | 0.668 | | 0.345 | 0.000 | *** | 0.365 | 0.000 | *** |
| spat. typ. rural peripheral | 0.145 | 0.007 | ** | 0.407 | 0.000 | *** | 0.408 | 0.000 | *** |
| age 0–17 | 0.321 | 0.392 | | -1.35 | 0.219 | | -7.03 | 1.000 | |
| age 18–29 | -0.628 | 0.000 | *** | -1.95 | 0.000 | *** | -1.96 | 0.000 | *** |
| age 30–39 | -0.310 | 0.000 | *** | -0.803 | 0.000 | *** | -0.831 | 0.000 | *** |
| *age 40–49* | | | | | | | | | |
| age 50–59 | -0.205 | 0.002 | ** | 0.426 | 0.000 | *** | 0.447 | 0.000 | *** |
| age 60–69 | -0.226 | 0.006 | ** | 0.755 | 0.000 | *** | 0.784 | 0.000 | *** |
| age 70–79 | -0.584 | 0.000 | *** | 0.363 | 0.013 | * | 0.397 | 0.007 | ** |
| age 80+ | -1.67 | 0.000 | *** | -0.864 | 0.000 | *** | -0.799 | 0.070 | * |
| *edu. none (yet)* | | | | | | | | | |
| edu. "Volks-/Hauptsch." | -0.212 | 0.000 | *** | 0.054 | 0.556 | | 0.096 | 0.030 | * |
| edu. "Mittlere Reife" | -0.088 | 0.090 | * | 0.048 | 0.580 | | 0.076 | 0.386 | |
| *edu. "Abitur"* | | | | | | | | | |
| edu. university degree | 0.170 | 0.001 | ** | 0.127 | 0.136 | | 0.114 | 0.186 | |
| edu. Other | -0.303 | 0.004 | ** | -0.216 | 0.227 | | -0.183 | 0.308 | |
| *sex male* | | | | | | | | | |
| sex female | -0.37 | 0.000 | *** | -0.389 | 0.000 | *** | -0.368 | 0.000 | *** |
| *household size 1* | | | | | | | | | |
| household size 2 | 0.533 | 0.000 | *** | 0.719 | 0.000 | *** | 0.753 | 0.000 | *** |
| household size 3 | 0.572 | 0.000 | *** | 0.565 | 0.000 | *** | 0.593 | 0.000 | *** |
| household size 4+ | 0.886 | 0.000 | *** | 0.906 | 0.000 | *** | 0.865 | 0.000 | *** |
| *occupation employed* | | | | | | | | | |
| occupation education | 0.178 | 0.085 | * | 0.304 | 0.370 | | 0.380 | 0.269 | |
| occupation domestic | -0.459 | 0.000 | *** | -0.053 | 0.686 | | -0.042 | 0.748 | |
| occupation retired | -0.464 | 0.000 | *** | -0.018 | 0.853 | | -0.015 | 0.879 | |
| occupation other | -0.526 | 0.000 | *** | -0.286 | 0.076 | * | -0.300 | 0.065 | * |
| eco. status very low | -0.393 | 0.000 | *** | -1.06 | 0.000 | *** | -1.08 | 0.000 | *** |
| eco. status low | -0.126 | 0.019 | * | -0.508 | 0.000 | *** | -0.506 | 0.000 | *** |
| *eco. status middle* | | | | | | | | | |
| eco. status high | 0.329 | 0.000 | *** | 0.515 | 0.000 | *** | 0.500 | 0.000 | *** |
| eco. status very high | 0.420 | 0.000 | *** | 0.667 | 0.000 | *** | 0.667 | 0.000 | *** |
| nest e-bike yes | 10.0 | 0.000 | *** | | | | ***/**/* = 0.1/1/10% | | |

category "metropolitan urban", utility for owning an e-bike is indeed positive in more peripheral regions. However after also taking into account gradient, there is a clear indication that this added utility peaks in rural areas close to cities and decreases again for very peripheral areas. The impact of spatial typology on c-bike utility appears negligible in magnitude.

**Table 4. Parameter values for the multivariate probit model.**

| Parameter | C-bike | | | E-bike | | |
|---|---|---|---|---|---|---|
| | Value | Rob. p-val. | Sig. | Value | Rob. p-val. | Sig. |
| constant | 1.138 | 0.000 | *** | -1.952 | 0.000 | *** |
| gradient | -3.373 | 0.000 | *** | 0.656 | 0.004 | ** |
| *spat. typ. metrop. urban* | | | | | | |
| spat. typ. regiop. urban | -0.087 | 0.000 | *** | 0.106 | 0.000 | *** |
| spat. typ. rural close to city | -0.014 | 0.597 | | 0.203 | 0.000 | *** |
| spat. typ. rural peripheral | 0.074 | 0.009 | ** | 0.097 | 0.007 | ** |
| age 0–17 | 0.103 | 0.542 | | -0.224 | 0.347 | |
| age 18–29 | -0.237 | 0.000 | *** | -0.645 | 0.000 | *** |
| age 30–39 | -0.138 | 0.001 | ** | -0.244 | 0.000 | *** |
| *age 40–49* | | | | | | |
| age 50–59 | -0.078 | 0.017 | * | 0.190 | 0.000 | *** |
| age 60–69 | -0.202 | 0.000 | *** | 0.360 | 0.000 | *** |
| age 70–79 | -0.389 | 0.000 | *** | 0.371 | 0.000 | *** |
| age 80+ | -0.972 | 0.000 | *** | 0.101 | 0.132 | |
| *edu. none (yet)* | | | | | | |
| edu. "Volks-/Hauptsch." | -0.191 | 0.000 | *** | 0.108 | 0.005 | ** |
| edu. "Mittlere Reife" | -0.116 | 0.000 | *** | 0.099 | 0.006 | ** |
| *edu. "Abitur"* | | | | | | |
| edu. university degree | 0.095 | 0.000 | *** | -0.053 | 0.138 | |
| edu. other | -0.183 | 0.001 | ** | 0.064 | 0.418 | |
| *sex male* | | | | | | |
| sex female | -0.186 | 0.000 | *** | -0.047 | 0.043 | ** |
| *household size 1* | | | | | | |
| household size 2 | 0.223 | 0.000 | *** | 0.261 | 0.000 | *** |
| household size 3 | 0.272 | 0.000 | *** | 0.132 | 0.003 | ** |
| household size 4+ | 0.447 | 0.000 | *** | 0.114 | 0.014 | * |
| *occupation employed* | | | | | | |
| occupation education | 0.022 | 0.680 | | 0.044 | 0.720 | |
| occupation domestic | -0.156 | 0.001 | ** | 0.175 | 0.001 | *** |
| occupation retired | -0.160 | 0.000 | *** | 0.148 | 0.000 | *** |
| occupation other | -0.192 | 0.000 | *** | 0.029 | 0.680 | |
| eco. status very low | -0.146 | 0.001 | ** | -0.234 | 0.002 | ** |
| eco. status low | -0.114 | 0.000 | *** | -0.156 | 0.000 | *** |
| *eco. Status middle* | | | | | | |
| eco. status high | 0.173 | 0.000 | *** | 0.055 | 0.045 | * |
| eco. status very high | 0.192 | 0.000 | *** | 0.186 | 0.000 | *** |
| R | -0.235 | 0.000 | *** | | ***/**/* = 0.1/1/10% | |

Age: As expected, the probit model describes a falling utility for c-bikes from the reference age group of 40–49 years onwards. More surprisingly, there is also a significant disutility for age 18–39 and only an insignificantly positive utility for age group 0–17 – albeit this category also expressing the effects of level of education and occupation for this youngest age group due to the interacting of these variables with age 18+. It therefore stands to reason that the higher rate of c-bike ownership among minors is more adequately explained by other factors, such as household size. For e-bikes, utility

**Table 5. Comparison of model properties.**

| Property | Nested logit | Multivariate probit |
|---|---|---|
| Number of parameters | 85 | 57 |
| Sample size | 30,000 | 30,000 |
| Null-log-likelihood | -41,588.8 | -41,588.8 |
| Log-likelihood | -21,694.0 | -22,333.4 |
| Adjusted $\rho^2$ [36] | 0.476 | 0.462 |

peaks around 60–79 years and decreases for both younger and older ages. According to the nested logit model, the utility of owning only a c-bike peaks around 40–49 years and has an additional upward tick for age 0–17, while owning only an e-bike is most attractive for age groups 50–79. We point out that in Germany, while riding so-called S-Pedelecs, which can reach speeds up to 45km/h, is subject to an age restriction of 16 years, the vast majority of e-bikes have no such restriction.

Education: According to the probit model, a higher level of education means a slightly positive utility for a c-bike, while in the case of e-bike ownership, only the slightly positive parameters for "Volks-/Hauptschule" and "Mittlere Reife" are significant. In the nested logit model, even fewer parameters are significant, with the results for bundle "only c-bike" mirroring the findings of the probit model.

Gender: According to the probit model, women show a lower utility for owning a c-bike compared to men, analogous to their slightly lower bicycle use [6]. For e-bikes, the impact of gender is much lower, albeit not zero. The nested logit model confirms this regarding c-bike ownership, however the two bundles containing e-bike are associated with a similar disutility to bundle "only c-bike".

Household size: The utility of any bundle increases with household size in the nested logit model. This was expected, as the probability that there is at least one bicycle in the household that can be shared increases as the number of people in the household rises. While probit mirrors this for c-bike, we find that e-bike utility peaks for two-person households. We hypothesise that this reflects the use of e-bikes primarily for leisure activities by couple households without children.

Occupation: Compared to the reference group of adult working people, housemen/-women, retirees, and other occupations show a significantly reduced utility for owning a c-bike. Owning an e-bike, on the other hand, is very clearly associated with a positive utility for retirees and housemen/-women. The nested logit model is less clear regarding the impact of occupation, other than that domestic and retired occupation goes along with a high disutility of owning only a c-bike.

Economic status: The higher the economic status, the greater the utilities of both a c-bike and an e-bike in the probit model. Likewise, in the nested logit model every combination of bicycle ownership also sees increased utility with higher economic status. This shows that bicycles are not generally used by low-income households as a substitute for a more expensive car, but instead are the result of a lifestyle choice.

R and nest parameters: The probit model's parameter $R$, i.e., the correlation of the error terms between a person's utility functions for the two different types of bicycle, can capture substitution effects, e.g., giving up a c-bike after purchasing an e-bike, as well as complementary effects. One conceivable complementary effect is that people with cycling-orientated attitudes (which are not explicitly included in our models and are therefore part of the error terms) have an additional positive utility for both a c-bike and an e-bike. $R$'s negative, highly significant value of -0.235 shows that the substitution effects clearly dominate and that the assumption of an independent distribution of the error terms is not tenable. This contrasts with findings by Ma et al. [28], who (between c-bikes and Chinese-style e-bikes) find a value of only +0.027. The nest parameter of 10.0 indicate very strong correlation between the alternatives in the "e-bike" nest. We note that when testing nesting structure 1 (Fig 4), the nest parameter of "no e-bike" came out as 1. This suggests that the decision of whether to own an e-bike is far more critical than the decision to own a c-bike.

## Suitability of model types and implications for policy and modelling practice

Both model types have advantages and disadvantages. Nested logit model coefficients can be interpreted as odds ratios and the model presented here achieves a higher model fit than the probit model even after accounting for the higher number of parameters. By modelling bundles, it can better depict their specific benefits for different groups of people, e.g., the phenomenon of e-bike-only owners among older senior citizens, while accounting for correlation between bundles using nest parameters. The probit model, on the other hand, can consider such correlations between the mobility tools of a bundle only with a global parameter $R$. However, the consideration of bundles represents a disadvantage for questions focussing on a single mobility tool, where the probit model can be interpreted more intuitively. This becomes even more relevant when more than two mobility tools are taken into account, as the number of bundles would grow exponentially. While it is possible to parameterise a nested logit model to allow for additive effects, this would forego the model's ability to capture bundle effects. The two models presented here therefore complement each other in terms of the findings and interpretations they allow.

Our model can be used as a predictive sub-model within a larger integrated transport model. For such use cases, the interpretability of model parameters is less relevant than predictive power. We therefore recommend using the nested logit model, as this variant achieved a higher model fit. We demonstrated that not only socio-demographic characteristics but also the variables of spatial typology and especially gradient significantly influence the utility of c-bike and e-bike ownership. Therefore, these variables should be included, especially when they vary substantially across the model area. Where data on c-bike and especially e-bike ownership is not available in Germany, our model can be used to gauge their magnitude, which is relevant for bicycle retailers and providers of bike sharing systems. For modelling efforts outside of Germany, our work can inform suitable model types and relevant explanatory variables. The model furthermore sheds light on the true causal relationships behind c-bike and e-bike ownership. For example, we were able to demonstrate that higher e-bike ownership rates in very rural areas identified in previous works are not primarily due to the urban structure itself, but rather due to more varied topography and older residents. With e-bikes already being viewed as a valuable mobility solution by the elderly and residents of hilly areas, targeted purchase incentives could further increase their uptake and consequently cycling among other groups.

## Limitations and further research needs

While our study has provided valuable insights into what factors influence c-bike and e-bike ownership, several limitations and avenues for future research remain to be explored. Bike-sharing systems were not considered, although they are a low-threshold option for getting to know e-bikes or substituting private e-bike ownership, particularly in urban areas. It was not possible to consider the price of bicycle types, which also would have made it possible to determine willingness to pay for other variables, due to a lack of data and the character of the MiD as a cross-sectional and revealed preferences survey (and thus a lack of variance in the purchase costs). It is conceivable that the variable gradient correlates with other factors such as local infrastructure quality or cycling culture, which were not analysed. Instead of gradient and spatial typology, which capture singular aspects of bicycle accessibility, future research could benefit from using a more holistic accessibility measure for c-bike and e-bike travel as an explanatory variable. As personal attitudes were not recorded in the MiD 2017, these could not be taken into account, although there is broad evidence in the literature for their relevance. The dynamic development of e-bike sales is probably largely due to changing attitudes and they are therefore of particular importance for predictive models. Since e-bike sales have already risen significantly again since 2017 [1], the present approach should be repeated in the form of a replication study once newer data becomes available.

## Conclusions

This study contributes to a better understanding of the choice of owning conventional and electric bicycles and suitable model types by estimating a nested logit and a multivariate probit model based on data from the MiD 2017 survey and

other sources. While the results of the multivariate probit model were more intuitively interpretably, the nested logit model achieved a higher model fit and could capture some bundle-specific effects. Regarding research question 1 (Which factors influence the choice to own a conventional and/or electric bicycle?) we can generally confirm the relationships known for the socio-demographic factors age, level of education, gender, household size, occupation, and economic status from the literature for the European context. Regarding research question 2 (What role does topography play in particular?) we find that while the utility for c-bike ownership decreases with average gradient around the residential location, this is not the case for electric bicycles. To our knowledge, we are the first to quantify this influence of the gradient of terrain near the residence location on conventional and electric bicycle ownership. Lastly, regarding research question 3 (How are the two choices interlinked?), the negative correlation of the error terms in the probit model suggests that unobserved substitution effects between the two types of bicycles outweigh unobserved complementary effects, providing the first evidence of its kind on this relationship. The adopted nesting structure and resulting nest parameter value of the nested logit model suggest that the choice to own a conventional bicycle is subordinate to the decision to own an electric bicycle.

Future surveys and analyses should take into account not only the influencing factors of gradient, spatial typology and socio-demographic variables but also personal attitudes in order to enable predictive ownership choice models. Building on this work, in subsequent research projects we will look at mode choice behaviour differentiated according to conventional and electric cycling and incorporate bicycle ownership into this.

## Author contributions

**Conceptualization:** Leonard Arning.

**Data curation:** Leonard Arning.

**Formal analysis:** Leonard Arning.

**Funding acquisition:** Heather Kaths.

**Investigation:** Leonard Arning.

**Methodology:** Leonard Arning.

**Project administration:** Leonard Arning.

**Resources:** Leonard Arning.

**Software:** Leonard Arning.

**Supervision:** Heather Kaths.

**Validation:** Leonard Arning.

**Visualization:** Leonard Arning.

**Writing – original draft:** Leonard Arning.

**Writing – review & editing:** Heather Kaths.

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
