## [Decision Letter · Decision Letter 0]

22 Jan 2025

PONE-D-24-51820Just another bike? Modelling the interdependence of conventional and electric bicycle ownership and the influence of topography using large-scale travel survey data from GermanyPLOS ONE

Dear Dr. Arning,

Thank you for submitting your manuscript to PLOS ONE. After careful consideration, we feel that it has merit but does not fully meet PLOS ONE’s publication criteria as it currently stands. Therefore, we invite you to submit a revised version of the manuscript that addresses the points raised during the review process.

We look forward to receiving your revised manuscript.

Kind regards,

Michał Suchanek, D.Sc.

Academic Editor

PLOS ONE

Journal Requirements:

“This research was supported via funding from the German Federal Ministry of Digital and Transport's Bicycle Traffic Endowed Professorship at the University of Wuppertal.”

5. For studies involving third-party data, we encourage authors to share any data specific to their analyses that they can legally distribute. PLOS recognizes, however, that authors may be using third-party data they do not have the rights to share. When third-party data cannot be publicly shared, authors must provide all information necessary for interested researchers to apply to gain access to the data. (https://journals.plos.org/plosone/s/data-availability#loc-acceptable-data-access-restrictions) 

6. We note that Figures 1 and 2 in your submission contain map images which may be copyrighted. All PLOS content is published under the Creative Commons Attribution License (CC BY 4.0), which means that the manuscript, images, and Supporting Information files will be freely available online, and any third party is permitted to access, download, copy, distribute, and use these materials in any way, even commercially, with proper attribution. For these reasons, we cannot publish previously copyrighted maps or satellite images created using proprietary data, such as Google software (Google Maps, Street View, and Earth). For more information, see our copyright guidelines: http://journals.plos.org/plosone/s/licenses-and-copyright.

1) You may seek permission from the original copyright holder of Figures 1 and 2 to publish the content specifically under the CC BY 4.0 license.  

2) If you are unable to obtain permission from the original copyright holder to publish these figures under the CC BY 4.0 license or if the copyright holder’s requirements are incompatible with the CC BY 4.0 license, please either i) remove the figure or ii) supply a replacement figure that complies with the CC BY 4.0 license. Please check copyright information on all replacement figures and update the figure caption with source information. If applicable, please specify in the figure caption text when a figure is similar but not identical to the original image and is therefore for illustrative purposes only.

**Additional Editor Comments:**

Thank you for submitting your manuscript to PLOS ONE. After consideration of the reviews provided by our expert reviewers, I have reached a decision of major revision for your manuscript. Below, I provide a summary of the key points raised by the reviewers to help guide your revisions:

Novelty and Contribution:

While the manuscript is well-written, the reviewers have noted that the degree of novelty and innovation is limited. To strengthen your submission, it is recommended to:

Clearly differentiate your study from existing works using the same dataset (e.g., Kohlrautz and Kuhnimhof).

Highlight the novel contributions of your study explicitly in the introduction, potentially through a dedicated "Main Contribution" section.

Practical Use Cases:

Reviewer 1 suggests that including practical use cases in the discussion section would add value to your study. These could include applications for transportation planners, business models for manufacturers, or other relevant domains such as simulations.

Clarity of Methods:

Reviewer 2 pointed out that the mathematical presentation of the nested logit model and multivariate probit model is not sufficiently clear. Adding more detailed equations and explanations would improve the clarity and rigor of your methodology section.

Figures and Data Visualization:

To enhance the readability and impact of your manuscript, Reviewer 2 recommends adding more comparative figures, particularly to illustrate key differences between conventional and electric bicycles. This would provide a stronger visual representation of your findings.

Formatting and Presentation:

Ensure consistency in formatting, especially in the references section.

Address any minor typographical errors to improve the overall presentation of the manuscript.

Data Accessibility:

Reviewer 1 noted that your manuscript does not fully adhere to the PLOS Data Availability Policy. Please ensure all underlying data are made fully accessible or clearly state any limitations.

I invite you to revise your manuscript in response to these comments. In preparing your revision, please provide a detailed response to each reviewer comment, indicating how you have addressed their concerns. If you choose not to address a specific comment, provide a clear and thorough justification. Please note that while the revisions address several substantial points, they do not necessarily guarantee acceptance, as the revised manuscript will undergo further evaluation.

Thank you for your submission.

Reviewers' comments:

Reviewer's Responses to Questions

**Comments to the Author**

1. Is the manuscript technically sound, and do the data support the conclusions?

Reviewer #1: Yes

Reviewer #2: Yes

2. Has the statistical analysis been performed appropriately and rigorously? 

Reviewer #1: Yes

Reviewer #2: N/A

3. Have the authors made all data underlying the findings in their manuscript fully available?

Reviewer #1: No

Reviewer #2: Yes

4. Is the manuscript presented in an intelligible fashion and written in standard English?

Reviewer #1: Yes

Reviewer #2: Yes

5. Review Comments to the Author

Reviewer #1: The paper apply basic models to survey based data (mobility in Germany) and two spatial data (degree of urbanisation and topography) to identify and analyse influential factors for e-bike ownership. The paper is well written, the methodology is clearly explained and the results are well presented.

However, it must be noted that the novelty and degree of innovation is very limited. Standard methods are applied to a data set that has already been analyzed in a similar way in many papers (some of which have been cited). The results clearly show and underline what one would already think influences e-bike ownership. Use cases on how the information can be used would be interesting and could be a part of the discussion. Apart from business models of bike manufacturers, there are certainly also use cases for transportation planners, simulations, etc. which one could elaborate.

Specific recommendation:

- Clearer differentiation from existing papers that use the same data set to investigate similar questions (e.g. Kohlrautz and Kuhnimhof)

- Clearly emphasize novelty and innovation, if necessary list the “Main Contribution” in the introduction

- Add specific use cases on how and where the results can be applied in discussion

Reviewer #2: Regarding this manuscript, I have the following suggestions for the author's reference, and I look forward to adding more readability to this study.

1.The formatting is not standardised in some places in the manuscript, such as the references.

2.There are only four figures in the manuscript, which is too few, and some relevant comparative figures of conventional bicycles and e-bikes can be added appropriately to make it easier for readers to understand.

3.Too few formulas to introduce the nested logit model and the multivariate probit model, not clear enough.

6. PLOS authors have the option to publish the peer review history of their article (what does this mean? ). If published, this will include your full peer review and any attached files.

**Do you want your identity to be public for this peer review?** For information about this choice, including consent withdrawal, please see our Privacy Policy .

Reviewer #1: No

Reviewer #2: No

---

## [Author Response · Author response to Decision Letter 1]

12 Feb 2025

We included a formatted version of the following response to reviewers (20250131_D_Response_to_reviewers.pdf) with this resubmission.

-----

Response to Reviewers

Paper Title: Just another bike? Modelling the interdependence of conventional and electric bicycle ownership and the influence of topography using large-scale travel survey data from Germany

Authors: Leonard Arning, Heather Kaths

Date of Reply: Jan 31st, 2025 (1st revision)

Dear Dr. Michał Suchanek, dear reviewers,

We would like to thank you for taking the time to review our manuscript and for your valuable comments. We have addressed your comments in the new manuscript as laid out below. Our responses are highlighted in dark blue.

We hope that our revisions address your concerns satisfactorily and that the updated manuscript meets your expectations. Please do not hesitate to let us know if further clarifications or adjustments are needed. Thank you again for your valuable feedback and for considering our manuscript.

Kind regards,

Leonard Arning

Reviewers’ and Editor’s comments with responses:

Journal Requirements:

RESPONSE: We have revised the formatting of our manuscript and the naming of the figure files to adhere to the PLOS ONE style requirements.

“This research was supported via funding from the German Federal Ministry of Digital and Transport's Bicycle Traffic Endowed Professorship at the University of Wuppertal.”

RESPONSE: We have clarified our financial disclosure and provide the text for the new statement in the new Cover Letter

RESPONSE: There was a technical issue in the online submission system which did not allow us to select the correct institution for funding. We reached out to PLOS customer care who replied with helpful instructions on how to proceed during the resubmission process, which we will follow.

RESPONSE: We updated our Data Availability statement to now include a more detailed description of the reasons and nature of the access restriction, conditions for data access and contact details.

5. For studies involving third-party data, we encourage authors to share any data specific to their analyses that they can legally distribute. PLOS recognizes, however, that authors may be using third-party data they do not have the rights to share. When third-party data cannot be publicly shared, authors must provide all information necessary for interested researchers to apply to gain access to the data. (https://journals.plos.org/plosone/s/data-availability#loc-acceptable-data-access-restrictions)

RESPONSE: We updated our Data Availability Statement to now include a more detailed description of the data ownership. We also clarified that we did not receive any special privileges and provided more precise contact information for researchers interested in using the data.

6. We note that Figures 1 and 2 in your submission contain map images which may be copyrighted. All PLOS content is published under the Creative Commons Attribution License (CC BY 4.0), which means that the manuscript, images, and Supporting Information files will be freely available online, and any third party is permitted to access, download, copy, distribute, and use these materials in any way, even commercially, with proper attribution. For these reasons, we cannot publish previously copyrighted maps or satellite images created using proprietary data, such as Google software (Google Maps, Street View, and Earth). For more information, see our copyright guidelines: http://journals.plos.org/plosone/s/licenses-and-copyright.

1) You may seek permission from the original copyright holder of Figures 1 and 2 to publish the content specifically under the CC BY 4.0 license.

2) If you are unable to obtain permission from the original copyright holder to publish these figures under the CC BY 4.0 license or if the copyright holder’s requirements are incompatible with the CC BY 4.0 license, please either i) remove the figure or ii) supply a replacement figure that complies with the CC BY 4.0 license. Please check copyright information on all replacement figures and update the figure caption with source information. If applicable, please specify in the figure caption text when a figure is similar but not identical to the original image and is therefore for illustrative purposes only.

RESPONSE: We address this concern in a separate file called “Image copyright declaration” submitted alongside the revised manuscript. The images were created by us without using copyright protected map images or prepublished figures. We only used data that we have permission to use. We clarified the data sources and data licenses used for the creation of these two figures in the figure captions and the references section.

Additional Editor Comments:

Thank you for submitting your manuscript to PLOS ONE. After consideration of the reviews provided by our expert reviewers, I have reached a decision of major revision for your manuscript. Below, I provide a summary of the key points raised by the reviewers to help guide your revisions:

Novelty and Contribution:

While the manuscript is well-written, the reviewers have noted that the degree of novelty and innovation is limited. To strengthen your submission, it is recommended to:

Clearly differentiate your study from existing works using the same dataset (e.g., Kohlrautz and Kuhnimhof).

Highlight the novel contributions of your study explicitly in the introduction, potentially through a dedicated "Main Contribution" section.

Practical Use Cases:

Reviewer 1 suggests that including practical use cases in the discussion section would add value to your study. These could include applications for transportation planners, business models for manufacturers, or other relevant domains such as simulations.

Clarity of Methods:

Reviewer 2 pointed out that the mathematical presentation of the nested logit model and multivariate probit model is not sufficiently clear. Adding more detailed equations and explanations would improve the clarity and rigor of your methodology section.

Figures and Data Visualization:

To enhance the readability and impact of your manuscript, Reviewer 2 recommends adding more comparative figures, particularly to illustrate key differences between conventional and electric bicycles. This would provide a stronger visual representation of your findings.

Formatting and Presentation:

Ensure consistency in formatting, especially in the references section.

Address any minor typographical errors to improve the overall presentation of the manuscript.

Data Accessibility:

Reviewer 1 noted that your manuscript does not fully adhere to the PLOS Data Availability Policy. Please ensure all underlying data are made fully accessible or clearly state any limitations.

I invite you to revise your manuscript in response to these comments. In preparing your revision, please provide a detailed response to each reviewer comment, indicating how you have addressed their concerns. If you choose not to address a specific comment, provide a clear and thorough justification. Please note that while the revisions address several substantial points, they do not necessarily guarantee acceptance, as the revised manuscript will undergo further evaluation.

Thank you for your submission.

RESPONSE: Thank you for summarizing the reviewers’ comments. We will address them individually below.

Reviewers' comments:

Reviewer's Responses to Questions

Comments to the Author

1. Is the manuscript technically sound, and do the data support the conclusions?

Reviewer #1: Yes

Reviewer #2: Yes

2. Has the statistical analysis been performed appropriately and rigorously?

Reviewer #1: Yes

Reviewer #2: N/A

3. Have the authors made all data underlying the findings in their manuscript fully available?

Reviewer #1: No

Reviewer #2: Yes

RESPONSE: See response regarding the updated Data Availability statement.

4. Is the manuscript presented in an intelligible fashion and written in standard English?

Reviewer #1: Yes

Reviewer #2: Yes

5. Review Comments to the Author

Reviewer #1: The paper apply basic models to survey based data (mobility in Germany) and two spatial data (degree of urbanisation and topography) to identify and analyse influential factors for e-bike ownership. The paper is well written, the methodology is clearly explained and the results are well presented.

However, it must be noted that the novelty and degree of innovation is very limited. Standard methods are applied to a data set that has already been analyzed in a similar way in many papers (some of which have been cited). The results clearly show and underline what one would already think influences e-bike ownership. Use cases on how the information can be used would be interesting and could be a part of the discussion. Apart from business models of bike manufacturers, there are certainly also use cases for transportation planners, simulations, etc. which one could elaborate.

Specific recommendation:

- Clearer differentiation from existing papers that use the same data set to investigate similar questi

---

## [Editor Report · Decision Letter 1]

19 Mar 2025

Just another bike? Modelling the interdependence of conventional and electric bicycle ownership and the influence of topography using large-scale travel survey data from Germany

PONE-D-24-51820R1

Dear Dr. Arning,

We’re pleased to inform you that your manuscript has been judged scientifically suitable for publication and will be formally accepted for publication once it meets all outstanding technical requirements.

Kind regards,

Michał Suchanek, D.Sc.

Academic Editor

PLOS ONE

Additional Editor Comments (optional):

Dear Authors,

I've reviewed the changes which you've made to the paper and find them to be a substantial improvement in line with reviewers' and mine recommendations. I am thus recommending an accept decision.
---

## [Editor Report · Acceptance letter]

PONE-D-24-51820R1

PLOS ONE

Dear Dr. Arning,

I'm pleased to inform you that your manuscript has been deemed suitable for publication in PLOS ONE. Congratulations! Your manuscript is now being handed over to our production team.

Kind regards,

on behalf of

Dr. Michał Suchanek

Academic Editor

PLOS ONE